# Recent Insight into the Role of Fibrosis in Nonalcoholic Steatohepatitis-Related Hepatocellular Carcinoma

**DOI:** 10.3390/ijms20071745

**Published:** 2019-04-09

**Authors:** Antonio Sircana, Elena Paschetta, Francesca Saba, Federica Molinaro, Giovanni Musso

**Affiliations:** 1Department of Cardiology, Azienda Ospedaliero Universitaria, 07100 Sassari, Italy; ant.sircana@gmail.com; 2HUMANITAS Gradenigo, University of Turin, 10132 Turin, Italy; elena.paschetta@alice.it (E.P.); fede.molinaro@gmail.com (F.M.); 3Department of Medical Sciences, Cittàdella Salute, University of Turin, 10126 Turin, Italy; francescasaba85@yahoo.it

**Keywords:** hepatocellular carcinoma, non-alcoholic steatohepatitis, fibrosis, hepatic stellate cells, extracellular matrix, carcinogenesis

## Abstract

Hepatocellular carcinoma (HCC) is one of the most widespread tumors in the world and its prognosis is poor because of lack of effective treatments. Epidemiological studies show that non-alcoholic steatohepatitis (NASH) and advanced fibrosis represent a relevant risk factors to the HCC development. However little is known of pathophysiological mechanisms linking liver fibrogenesis to HCC in NASH. Recent advances in scientific research allowed to discover some mechanisms that may represent potential therapeutic targets. These include the integrin signaling, hepatic stellate cells (HSCs) activation, Hedgehog signaling and alteration of immune system. In the near future, knowledge of fibrosis-dependent carcinogenic mechanisms, will help optimize antifibrotic therapies as an approach to prevent and treat HCC in patients with NASH and advanced fibrosis.

## 1. Introduction

As declared by the WHO (World Health Organization) Global Hepatitis Report [1], hepatocellular carcinoma (HCC) is the fifth most common cancer in the world and the second most common cause of death related to cancer in the last years. The main risk factors are chronic HBV and HCV infection, NAFLD/NASH, alcoholic hepatitis, and any disease leading to cirrhosis.

Non-alcoholic fatty liver disease (NAFLD), with global prevalence of 25% [2], is the most common cause of chronic liver disease. Its clinical spectrum ranges from simple hepatic steatosis (in the absence of secondary causes) to non-alcoholic steatohepatitis (NASH), a more aggressive form with inflammation, hepatocyte injury and varying degrees of fibrosis [3,4].

NASH prevalence has increased exponentially over the years and probably in the next decades it will be the leading cause of liver transplantation in the western industrialized countries. Epidemiological data suggest that HCC attributed to viral infection is in decline, while cases of cryptogenic or NASH-related HCC are significantly increased [5]. Across studies, the risk of HCC is not uniform among patients with NASH, and it ranges between 2.4% and 12.8%. If on the one hand it has been shown that patients with NASH can develop HCC in the absence of fibrosis or cirrhosis [6,7], on the other, it is shown that fibrosis plays a crucial role in causing HCC and its presence is correlated with poor prognosis [8,9]. A metanalisis of 17 cohort studies found a greater risk of HCC in NASH-cirrhosis cohorts than in NAFLD or NASH without fibrosis/cirrhosis [10]. HCC development is considered as result of different environmental risk factors that engage distinct genetic, epigenetic, and chromosomal alterations. It originates from chronic liver injury through a complex multistep process that involves several pathogenic mechanisms that contribute to carcinogenesis [11]. In humans, it is very complicated to identify the different molecular mechanisms underlying the pathogenesis of NASH and to understand how this evolves towards HCC. To overcome this problem, reproducible and representative preclinical models that are susceptible to genetic and functional analysis are used. Many NASH mouse models have been described, but some of them do not closely reflect human disease, thus hindering the efforts to definitively connect the various pathophysiological processes [12].

In this review we will discuss the latest findings on the pathophysiological mechanisms linking liver fibrogenesis to HCC in NASH and their potential therapeutic targets.

## 2. Pathophysiological Mechanisms of NAFLD Progression

NASH differs from simple steatosis by the presence of hepatocyte death, inflammation and various degrees of fibrosis. Cell death and inflammation play a leading role in disease progression, through HSCs’ activation and the subsequent fibrosis. Lately, scientists put forward the idea (“the multiple parallel hits hypothesis”) that progression from simple steatosis to NASH is the result of several disorders acting in parallel, including genetic predisposition, altered lipid metabolism, lipotoxicity, oxidative and endoplasmic reticulum stress, mitochondrial dysfunction, abnormal production of cytokines and adipokines, gut dysbiosis, and translocation of gut-derived LPS [13]. On this theory, hepatic inflammation constitute the “primum movens” of fibrosis progression in NASH. Genetic susceptibility and poor eating habits predispose to the development of insulin resistance and hepatic steatosis. In this context, lipotoxic metabolites of saturated fatty acids (SFA) can cause lipotoxicity, process that leads to cellular damage through excessive oxidative stress [14,15]. Damaged hepatocytes release DAMPs (damage-endogenous-associated molecular patterns) that activate pro-inflammatory signaling pathways via toll-like receptors (TLRs). Subsequent activation of Kupffer cells (KC) and inflammasome promote the massive release of pro-inflammatory, pro-fibrogenic cytokines and ligands. HSCs are then stimulated to produce high amount of extra-cellular matrix leading to progressive fibrosis [15]. KCs activation favors a pro-inflammatory microenvironment that triggers an adaptive immune response Th17-mediated. Moreover, chronic portal inflammatory infiltrate boosts a ductular reaction (DR) and hepatic progenitor cells (HPC) recruitment. All of these factors encourage progressive fibrosis that constitutes an imbalance between tissue injury and repair secondary to influence of vary inflammatory cells [16].

Acute inflammation constitutes a useful reaction to achieve tissues recovery by promoting regeneration. Conversely, chronic inflammation is maladaptive and provides a fertile soil to the development of liver fibrosis and HCC. Chronic injury triggers secretion of significant amounts of proinflammatory molecules including IL-1, IL-6, TNF-α, lymphotoxin-β that facilitate HCC development [17]. Activation of inflammatory signaling pathways and enhanced secretion of inflammatory molecules increase release of reactive oxygen species (ROS) by hepatocites. ROS can increase the tumor risk by mechanisms including DNA damage [11] and inhibition on immunosurveillance [18].

Although there is an established relationship between inflammation and fibrogenesis, it looks like that some inflammatory pathways selectively impact the tumorigenesis without affecting fibrosis. For instance, lymphotoxin-β and neutrophils, promote the development of HCC but have no known role in hepatic fibrogenesis [19]. In diethylnitrosamine (DEN) mice model, obesity and chronic inflammation enhanced production of IL-6 and TNF and promoted HCC development through activation of the oncogenic transcription factor STAT3 [20]. In an elegant study, Grohmann et al. demonstrated how obesity-associated hepatic oxidative stress can independently contribute to the pathogenesis of NASH, fibrosis, and HCC via STAT-1 and STAT-3 signaling [21]. On the contrary, some inflammatory cells such as macrophages may promote both fibrosis and HCC [17]. Activation of TLR4 by gut-derived lipopolysaccharide (LPS) promotes both as well [22]. In the light of the above, it may be concluded that inflammation promotes hepatocarcinogenesis through fibrosis-dependent and -independent pathways.

## 3. Fibrosis-Dependent Hepatocarcinogenesis

Fibrogenesis is a multi-cellular response that occurs whenever there is hepatic damage with hepatocellular death. Acute liver injury triggers the inflammatory and fibrogenic cascade with activation of the HSCs that constitute the main source of extracellular matrix (ECM) rich in collagen Ⅰ and Ⅲ [23]. The aim is to restore the architecture and function of the organ after serious damage. In fact, inflammatory signals promote hepatic regeneration, inflammatory cells provide for removal of cellular debris, while fibrosis enable the mechanical stability [24].

However, wound healing responses grow into harmful ones when the underlying trigger cannot be removed and the hepatocellular death becomes chronic, causing chronic inflammation and the development of progressive liver fibrosis which distorts the hepatic and vascular architecture [25].

Several stimuli are directly hepatocarcinogenic and the inflammation–fibrosis–cirrhosis–HCC paradigm does not provide a causal link between fibrosis and HCC. In fact, fibrosis is just one component that can be difficult to separate from other carcinogenic insults.

Because most of the data is associative rather than causal, it could be legitimate to suppose that fibrosis could only be a spectator in the process of carcinogenesis. But in the last few years, researchers have sought to create representative models that reflect the response of the human liver to injury in order to provide a mechanistic link between these two phenomena.

Potential mechanisms of fibrosis-dependent hepatocarcinogenesis include enhanced integrin signaling by ECM; paracrine crosstalk between HSCs, hepatocytes and the ECM; augmented stromal stiffness; hypoxia; imbalance between matrix metalloproteinases and tissue inhibitor of metalloproteases; excessive activation of Hedgehog pathway signaling; autophagy, hepatic progenitor cells recruitment and dysregulation of the immune system (Figure 1).

### 3.1. Extracellular Matrix, Hepatic Stellate Cells, and Integrins

In NASH, chronic hepatocellular damage and inflammation cause the activation of regenerative pathways and the proliferation of fibrogenic cells creating a micro-environment favorable to cellular survival and proliferation [26]. HSCs are ‘quiescent’ liver vitamin A-storing cells located in the perisinusoidal space. During the process of liver injury, the progressive release of pro-inflammatory molecules (platelet derived growth factor (PDGF), transforming growth factors-β (TGF-β), tumor necrosis factor-α (TNF-α), interleukin (IL)-1 and several chemokine) fosters HSCs activation and successive differentiation into contractile and fibrogenic myofibroblasts, which are characterized by upregulation of mesenchymal markers (α-smooth muscle actin (α-SMA, ACTA2), desmin (DES), and collagen α1(I)) [27]. Activated HSCs produce large amounts of fibrillar type I and III collagens, but also fibronectins, laminins, and fibrinogen, creating tight and highly crosslinked collagen bundles. An excessive production of ECM and reduction of ECM turnover is the typical characteristic of liver fibrosis. Changes in the composition and structure of the ECM have been shown to provoke cellular responses through the integrin family of transmembrane receptors, which play a crucial role in the activation of TGF-β and fibrogenic response [28,29]. Integrins modulate proliferation, differentiation and survival through the activation of Hedgehog signaling [30] and other intracellular specifics pathways, including protein kinase C, phosphatidylinositol 3-kinase (PI3K) and mitogen-activated protein kinases (MAPK) [31]. Integrins may also be involved in signal transduction during angiogenesis by stimulating the intracellular signaling molecules. The integrin expression pattern are different and up-regulated in primary and metastatic HCC tissues compared to normal hepatocytes. In human HCC cell lines, integrins α1β1 and α2β1 inhibition reduces migration induced by profibrotic growth factors including TGF-β1, epidermal growth factor (EGF) and fibroblast growth factor (FGF) [32]. However there are disagreements about the role of these receptors in the onset and progression of HCC [33]. Recently, Zheng et al. showed that collagen I promotes HCC cells proliferation by regulating the β1/FAK integrin pathway in murine models of NAFLD/NASH [34].

ECM proteins interact even with DDR proteins (discoidin domain receptor). Of these, DDR2 is a tyrosine kinase of the type I collagen that promotes epithelial-mesenchymal transition (EMT), an important mechanism that favors the malignant transformation of epithelial cells [35]. In animal model of pulmonary fibrosis, treatment with an antisense oligonucleotides (ASO) or with DDR2 inhibitors, prevents myofibroblast activation and angiogenesis [36].

### 3.2. Matrix Stiffness

In liver fibrosis, excessive collagen deposition increases the stiffness of the ECM, which in turn promotes the further increase in collagen secretion by HSCs [37]. A meta-analysis of 17 prospective cohort studies including 7058 patients found that increasing liver stiffness (measured using transient elastography) was associated with higher risk of HCC development [38].

Higher matrix stiffness, caused by copious matrix protein deposition and crosslinking, plays an important role in cell growth, proliferation, motility and tumor metastasis in several tissues [39,40].

Experimental data suggest that matrix stiffness might modulate HCC cells proliferation, invasion, angiogenesis through several pathways including extracellular signal-regulated kinase (ERK), protein kinase B (PKB/Akt), signal transducer and activator of transcription 3 (STAT3), integrin β1/GSK-3β/β-catenin [41] β1-integrin/focal adhesion kinase (FAK) [42], and β1-integrin/phosphatidylinositol-3- kinase (PI3K)/Akt signaling pathways [43]. Recent data indicate that ECM stiffness could promote tumorigenesis through activation of Hippo-YAP/TAZ signaling pathway, a critical regulator of cell proliferation, differentiation and apoptosis [19].

Recently, You et al., showed that matrix stiffness could take an active part in the process of stemness regulation through integrin β1/Akt/mTOR/SOX2 signaling pathway [44].

### 3.3. Matrix Metalloproteinases and Tissue Inhibitor of Metalloproteases

Matrix metalloproteinases (MMPs), (calcium-dependent enzymes secreted mainly by HSCs but also by macrophages) and their endogenous inhibitors, tissue inhibitor of metalloproteases (TIMPs) regulate the ECM turnover [45]. In animal models, an imbalance between MMPs and TIMPs concentrations is associated not only with an alteration of ECM homeostasis, but also with alterations of various biological activities that increase the risk of developing rheumatoid arthritis, atherosclerosis, nephritis, fibrosis, cirrhosis and cancer [46]. In fact, MMPs degrade the stroma, can remove the extracellular receptors on the cell surface and are also involved in the turnover of non-matrix substrates: MMPs liberate growth factors, ligands and citokines from the ECM. In addition, they are involved in the metabolism of macromolecules (proteins, lipids, carbohydrates) and regulation of growth and cellular differentiation [47].

Animal and human data suggest that MMPs and TIMPs are implicated in phenomena such as angiogenesis, proliferation of HSCs and progression of hepatocytes from dysplasia to HCC [48]. Indeed, TIMP-1 inhibits the tumor apoptosis via SDF-1/PI3K/AKT signaling [49]. MMP-1 production enhances proliferation, invasion and fibrosis in NASH. Moreover, serum MMP-1 levels reflect disease activity and may be used as a potential biomarker for monitoring the progression of disease [50]. MMP1 and MMP7 promote re-epithelialization and smooth muscle cells de-differentiation via PAR-1; MMP8 activates HSCs; MMP3 promotes the HGF-induced invasion of human HCC; MMP-19 is involved in TGF-β signaling; MMP-25 attenuates alpha-1 proteinase inhibitor facilitating the migration. In hepatoma cells, expression of MT1-MMP, MMP-2, and MMP-9 facilitates stromal invasion [45]. These data indicate that the imbalance between MMPs expression and their endogenous inhibitors could play a key role in HCC development. Researches investigating potential drug therapies targeting MMPs and TIMPs in HCC are currently ongoing.

### 3.4. Hypoxia

Abundant ECM deposition and subversion of normal architecture during the fibrogenic process creates altered blood flow that reduces metabolic exchange of oxygen in the liver parenchyma [51]. Hypoxia leads to activaction of specific signalling pathways [52] (PI3K-Akt and MAPK pathways) and upregulation of angiogenic factors including vascular endothelial growth factor (VEGF) and [53] hypoxia-inducible factor 1 (HIF-1) [54]. These facilitates tumor growth, angiogenesis, EMT and metastasis [55]. High levels of HIF-1 and VEGF correlate with the aggressiveness of HCC and with worst prognosis [56].

Tuftelin1 (TUFT1) is an acidic protein expressed in several tissues and exerts multifunctional roles [57]. New study in human with HCC found that hypoxia enhances TUFT1 expression through HIF-1α/miR-671-5p/TUFT1/AKT signaling pathway [58]. TUFT1 furthered HCC cell growth and metastasis in vitro and in vivo by activating the Ca2 /PI3K/AKT pathway. In this study TUFT1 knockdown minimized the promoting effects of hypoxia on tumor growth and metastasis, suggesting that TUFT1 may represent a new potential therapeutic target for HCC treatment.

### 3.5. Hedgehog Signaling

During embryogenesis, the Hedgehog (Hh) pathway represents an essential signaling mechanism that modulates many aspects of cell differentiation and tissue development [59]. In healthy adult liver, there is a low expression of Hh ligand and high expression of Hh ligand antagonist, so there is almost no Hh activity [60]. Liver injury, in contrast, is associated with elevated Hh signaling that stimulates liver regeneration [61]. Several studies show that in HCC there is an excessive activation of Hh signaling and this promotes proliferation, migration and invasion of HCC cells [62]. A recent study claims there is interplay between Sonic Hedgehog (Shh) and TGF-β1 in hepatic inflammatory reactions. Secreted Shh may involve activation of TGF-β1 and subsequent activation of HSCs, which together promote the progression of human NASH [63]. In Mdr2^-^/^-^ mice with chronic liver fibrosis, Philips et al. studied the role of the Hh pathway in hepatocarcinogenesis [64]. In this work, Hh pathway activation promotes both liver fibrosis and hepatocarcinogenesis while inhibition of Hh signaling reverses both processes. Authors suggest that the carcinogenic effect of Hh could be mediated by augmented myofibroblast activation and fibrosis [64].

### 3.6. Hepatic Progenitor Cells

Hepatic progenitor cells (HPCs) are resident stem cells located at the level of Canals of Hering and when activated promote tissue turnover and liver regeneration [65]. Chronic hepatocyte injury is associated with HPCs activation and enhancement of several pathways identified in liver cancer, including Hg, canonical Wnt signaling and Notch [66]. Proliferation and differentiation of HPCs, depend on the up-regulation of these pathways [67]. Studies in NASH patients claimed that HPCs activation and the expansion of ductular reaction (DR) were independently correlated with progressive fibrosis both in adult and children [68,69]. During hepatic necrosis, proliferating HPCs augment their expression of profibrogenic factors while DR cells produce PDGF, TGF-β, Sonic Hg and activate HSCs [65]. These findings support the idea that HPCs activation could contribute to the initiation and sustain HCC.

### 3.7. Autophagy

Autophagy is an evolutionarily conserved cellular process for lysosomal degradation of damaged cell components. Cellular organelles, lipid droplets, large protein aggregates are sequestered and degradated, while amino acids generated are used for producing energy [70]. Autophagy is a complex dynamic process regulated by several signaling pathways involved in cellular proliferation and apoptosis such as PI3K/Akt/mTOR and AMPK pathway [71]. In NASH, activated HSCs develop autophagic activity as a mechanism of lipid droplet degradation from which obtain energy support for their activation. In fact, treatment with an autophagy inhibitor prevents HSCs activation in vitro while reduces lipid droplet degradation [72]. This suggests that autophagy could be a target for the treatment of NASH and fibrosis. In rats, curcumin treatment leads to protection against toxin-induced HCC through induction of autophagic pathway and inhibition of apoptosis [73].

However the role of autophagy on HCC development is controversial. A study in autophagy-deficient mice with mosaic deletion of Atg5, showed the ‘double-edged sword’ of autophagy, since it is important for suppression of tumorigenesis in the liver but at the same time it promotes tumor progression because of accumulation of harmful protein aggregated [74]. Conversely, Sun et al. found that autophagy-deficient Kupffer cells promote liver inflammation, fibrosis, and HCC by enhancing mitochondrial ROS- NF-kB-IL1α/β pathways [75]. These data recommend that targeting autophagy for the treatment of NASH-fibrosis and HCC should require cell-specific autophagy inhibitors.

### 3.8. Dysregulation of the Immune System

The innate and adaptive immune system are essential for the identification and suppression of transformed cells. In NASH, liver injury stimulates the activation of several types of immune cells [76].

Kupffer cells (KCs) could play a important role in initiation and progression of inflammation and fibrosis. Liver biopsies of patients with NAFLD found a higher expansion of KCs than in controls. This phenomenon precedes the recruitment of other inflammatory cells, in fact macrophage infiltration occurs in more advanced inflammatory stages of the disease [77]. In the early phase of NAFLD, pathogen-associated molecular patterns (PAMPs), changes in gut microbiota, increase intestinal traslocation of bacteria and toxins can activate KCs [78], which secrete TGF-β, TNF-α, pro-inflammatory cytokines such as CCL2, ROS and activate inflammasomes [76]. Activation of NLRP3 fosters secretion of IL-1β, IL-18, and IL-6 which in turn promote disease progression and HCC development [79]. Furthermore, damaged hepatocytes are be able to activate KCs through cell stress pathways such as c-Jun N-terminal kinase (JNK) and release of DAMPs factors which promote inflammation via TNF-α, nuclear factor (NF)-κB and TLR signaling activation [78,80]. In addition, in condition of hypoxia KCs can be activated by HIF-1α [81]. Thus far, the role of KC in HCC is not fully clarified and is being investigated.

The role of dendritic cells (DC) in NASH is not clear. Existing experimental data are inconsistent perhaps because differences in the NASH models utilized or the diets administered in the studies [82].

The role of neutrophils is not also entirely clarified. However increase in liver neutrophils has been reported in human with NASH [83], and the degree of infiltration is correlated with severity of disease [84]. Neutrophils could contribute to NASH progression and carcinogenesis via Myeloperoxidase (MPO)-related mechanisms [84].

Natural killer (NK) cells are significantly elevated in NASH livers, compared to normal healthy control. Their activation during injury may be due to the higher levels of several cytokines including IL-12, IL-18, and IFN-γ [85]. In the early phases of disease, NK cells act against fibrosis development through IFNγ and by inducing apoptosis of HSCs via TRAIL and FasL, while in the late stages, NK cell function is compromised leading to further increases the ECM deposition promoting HCC development [86]. NK cells play a critical role in the immune surveillance of liver tumors [87]. Several mechanisms has been proposed to explain the decrease in the NK cell functions that are associated with advance fibrosis and HCC. These include TGFβ-mediated inhibition, phagoctyosis of NK cells by HSCs, inability to make target-cell contact and the dysregulation of activating ligands [88]. However, further studies are required to clarify the role of NK cells in the carcinogenesis process during fibrosis.

NKT cells increase in human NASH with advanced fibrosis but their role in hepatocarcinogenesis is still unclear. They can secrete IL-4, IFN-γ, and TNF-α [89] as well as promote steatosis through signaling via the lymphotoxin-like inducible protein LIGHT, activate HSCs, enhance Hh signaling then fibrosis [76]. The results of experimental data in HCC are in contrast. Indeed, NKT cells may both play an anti-tumor role and promote tumor tolerance [76].

In a choline-deficient high-fat diet mice model, Wolf et al., described intrahepatic activation of CD8^+^ T cells, NKT cells, and inflammatory cytokines, similarly to NASH patients [83]. CD8^+^ T cells and NKT cells synergistically induced steatosis, NASH and HCC development. NKT cells promote steatosis via lymphotoxin (LT)-like inducible protein LIGHT, while CD8(+) T cells cause liver damage in a LTβR-independent manner. CD8+ cytotoxic T lymphocytes kill their target cells not only through their two mayor cytotoxic mechanisms (perforin/granzyme-mediated, and Fas ligand (FasL)-mediated) [90], but also by secretion of IFN-γ and TNF-α [91]. However, depletion of CD8+ T lymphocytes in different experimental mice models produced varying results on onset and progression of HCC [92,93].

Previous studies reported that CD4+ T lymphocytes are capable to inhibit HCC initiation and favor tumor regression through the expression of chemokines [94]. Study in mouse models of NAFLD and HCC, found that dysregulation of lipid metabolism induces a ROS-mediated selective loss of intrahepatic CD4+ T but not of CD8+ T cells, leading to tumorigenesis [18]. An analysis of 547 patients with HCC showed that progressive loss of CD4^+^ cytotoxic T cells was significantly correlated with an advanced stage of disease and poor prognosis [95].

Regulatory T cells (Treg) recruitment may impair the function of CD8^+^ T cells and promote cancer progression. In a DEN-induced HCC mouse model, TGF-β promoted Treg cell differentiation, and this was identified as a major inhibitory mechanism of CD8+ T cells [96]. An increased recruitment of Treg cells correlated with poor prognosis [97].

In humans, NASH with advanced fibrosis is associated with high circulating IgA^+^ cells levels that build up in fibrotic liver [98]. These cells can interfere with activation of cytotoxic CD8^+^ T lymphocytes through programmed death ligand 1 (PD-L1) and interleukin-10, and promote HCC development [99]. In mice, PD-L1 blockade induces cytotoxic T-lymphocyte-mediated regression of established HCC [93].

Although the mechanisms are not yet well-known, we can state that immune system may play a ‘dual and opposite role’ in the development and progression of HCC. However, extensive studies are needed.

### 3.9. Crosstalk between NASH and Hepatocellular Carcinoma

HSCs, fibroblasts, immune cells, endothelial and mesenchymal stem cells as well as cytokines, growth factors and ECM constitute the liver tumor microenvironment. The results of numerous studies provide evidence that the cross-talk between tumor cells and their surrounding microenvironment is essential for cell growth, proliferation, EMT and metastasis [100] (Table 1).

Activated HSCs secrete several molecules including PDGF-B and PDGF-C, TGF-α, TGF-β, EGF, VEGF, angiopoietin-1 and -2, hepatocyte growth factor (HGF), stromal-derived factor-1alpha (SDF-1), Wnt ligands, interleukin-6 (IL-6) and epimorphin (EPM) [101]. These mediators are important pro-angiogenic, proliferative, and regenerative citokines that create a favorable microenvironment facilitating tumor initiation and progression [56]. In fibrotic liver, these molecules are passively sequestered by the ECM favoring the bidirectional interaction between endothelial, stromal cells and hepatocytes in an autocrine/paracrine manner [102]. Therefore, HSCs can be the target of the molecules produced by themselves. Activated HSCs show increased expression of different receptors for soluble cytokines, including PDGF that is the most potent proliferative cytokine [103]. Generally, expression of PDGF receptors by HSCs is low but drastically increases during inflammation, NAFLD and NASH [104]. Induction of β-PDGF receptors leads to activation of the Fas-MAPK pathway and release of intracellular calcium ions that activate PKC family members then activation of a more contractile and fibrogenic phenotype of HSC [105]. In mice model, hepatic over-expression of PDGF-C induces changes in gene expression, inflammation, progressive fibrosis, neoangiogenesis, and dysplasia. In patients with NASH, Wright et al., found a important correlation between PDGF-CC levels in liver and severity of disease [106]. Thus, PDGF-C could be more crucial in modulating the microenvironment to promote HCC development in a paracrine manner than in promoting direct carcinogenic effects on hepatocytes. In vitro, EPM promotes HCC cells invasion and metastasis by activating MMP-9 expression through the FAK-ERK pathway [107].

New data suggest that HSCs could promote HCC progression through the production of IL-1β, via a mechanism that seems to be dependent on PKR activation [108].

Cancer-associated fibroblasts (CAFs) are the most abundant cell type of the tumor stroma and are similar in morphology and molecular expression profiles to the myofibroblasts (HSCs) that are activated during the wound repair process [109]. However origin of the CAFs in HCC is obscure. Indeed, they can originate from HSCs, migrated bone-marrow stem cell and EMT [110]. Thus far, there is little evidence that HSCs and CAFs drive the malignant transformation of hepatocytes, but it is established that these stromal cells create a microenvironment that supports the growth of dysplastic hepatocytes and HCC [19]. Studies in vitro suggest that HGF could be a mediator of tumor–stromal interactions through which CAFs regulate the proliferation and invasion of HCC cells [111].

MMP-2 together with MMP-9 has a fundamental role on degrading type IV collagen that is the most abundant component of ECM. Feng et al., found that co-cultures of vascular endothelial and HCC HepG2 cells increased expression of MMP-2 and MMP-9 which enhanced the invasion ability of the HepG2 cells [112].

Angiogenesis provides a source of oxygen, nutrients and is indispensable for tumor growth and metastasis [113]. Both in phase of tumor development and progression, several molecular pathways are involved in the induction of angiogenesis and in the preservation of metastasis supporting vascular networks. VEGF represents a fundamental element that controls most of mechanisms of tumor-induced angiogenesis, because stimulates vascular sprouting, tip cells formation, sprout elongation and lumen formation [114].

In this context, tumour cells may activate Treg cells and promote immune tolerance by inhibiting the anti-tumorigenic effects of NK and CD8^+^ T cells [115].

## 4. Therapeutic Perspective

Although numerous drugs have been indagated, none of them have been validated in phase III.

Trials and so far, there is no medicament authorized by regulatory authorities for management of NASH. However, several molecules were shown to be effective in preclinical studies and some of these are currently being examined in humans.

TGF-β induces a complex modulation of gene expression because it can use several extracellular signals and adhesion molecules [116]. It promotes liver inflammation, activation of HCSs then fibrosis, EMT and growth of HCC [117]. TGF-β is regarded as pivotal molecule in HCC tumorigenesis since it, secreted by HSCs or by transformed hepatocites, may inhibits NK cell functions [118], and may control the secretion of other cytokines [119]. In human HCC cell lines, treatment with TGF-β inhibitor LY2109761 stops migration and invasion by upregulating E-cadherin [120], inhibits neo-angiogenesis, reduces tumor growth and metastasis of HCC cells by inhibiting CAFs proliferation [121]. Galunisertib (LY2157299) is another promising antifibroticTGF-β inhibitor that inhibits SMAD2 phosphorylation and blocks the collagens deposition promoting their degradation [122]. Large RCT in humans are needed.

Chemokines, regulate many functions of hepatocytes, endothelial cells, HCSs, and circulating immune cells. Interactions between C-C chemokine receptors (CCR2, CCR5,CCL2,CCL5,) and their ligands, promote fibrogenesis by HSCs activation and macrophage recruitment in the liver [123]. In preclinical models of NASH, Cenicriviroc (CCR2/CCR5 antagonist) improved hepatic inflammation and fibrosis [124]. These results have been confirmed in the recent phase 2b CENTAUR study [125]. A phase III trial to evaluate the efficacy of this drug is ongoing.

Sorafenib is multikinase inhibitor of VEGFR and PDGFR approved for the treatment of HCC where increases overall survival compared to placebo [126]. In NASH rodent models, sorafenib treatment decreased inflammation, angiogenesis, HSCs activation, collagen deposition, and hepatic fibrosis [127]. It may be considered for the treatment of NASH in humans. Further studies are warranted.

Inhibition of collagen synthesis may represent a potential therapy for fibrotic liver diseases. Recent study in rats found that cationic lipid nanoparticles loaded with small interfering RNA to the procollagen α1(I) gene administration, provoked specific inhibition of type I collagen synthesis without visible side effects [128]. So far, no data on human are avaiable.

In recent years, results of phase 2-3 trials found that treatment with integrin inhibitors was ineffective for the treatment of various cancers including HCC [129]. However, it has recently been discovered that αv integrins play a key role in the fibrogenesis in the liver, skin, kidney, and lung, although many mechanisms are still unknown [130]. Many Phase 1-2 trials are underway to evaluate the efficacy of integrin inhibitors in reversing the fibrosis process (ClinicalTrials.gov).

Hh dysregulation represent a novel mechanism for hepatic fibrosis and hepatocarcinogenesis, and may considered as potential therapeutic target for patients with NASH or HCC. Recently Hh-inhibitor LDE225 was approved for the treatment of basocellular carcinoma [131]. Phase 1 study to test the safety and determine the maximum safe dose of LDE225 in patients with HCC is ongoing (NCT02151864).

In fibrotic livers, the immunosuppressive function of the fibrosis-stimulated IgA^+^ cells probably depends on the expression of PD-L1 and IL-10, which may promote CTL dysfunction [132]. Programmed death 1 (PD-1) inhibitors reverse CD8^+^ T cells dysfunction and can therefore represent a treatment option for patients with advanced HCC. Nivolumab is a PD-1 inhibitor approved for the treatment of several malignancy including melanoma, non-small cell lung cancer and renal carcinoma. In a phase 1/2 trial, Nivolumab showed a manageable safety profile and no side effects were observed in patients with advanced HCC [133]. Several phase 2/3 Trial are ongoing (ClinicalTrial.gov).

## 5. Conclusions and Discussion

Growing evidence suggest a mechanistic link between fibrotic microenvironment and the HCC development. However, the lack of representative animal models is hampering the efforts to understand the pathophysiological mechanisms in NASH-related HCC. NASH is a complex and extremely heterogeneous metabolic disease and despite several mouse models can mimic disease; rarely, they replicate the pathogenic sequence of human NASH-HCC. These features can explain the conflicting results among studies using different animal models.

There are 3 main categories of murine NASH models: diet models, toxins/diet-based models, and genetic/diet models [12]. For instance, a methionine/choline-deficient (MCD) diet produces histological features of NASH, but rodents treated with this diet do not develop insulin resistance [134]. A western diet (WD), induces NASH, obesity and insulin resistance but disease does not progress to advanced fibrosis [135]. Streptozotocin (STZ) and diethylnitrosamine (DEN) + HFD models, develop obesity, insulin resistance, type 2 diabetes, mild fibrosis and HCC. However an independent carcinogenic effect of toxins cannot be excluded [12]. Tsuchida et al. studied a new mice model where mice were treated with WD and weekly dosing of carbon tetrachloride (CCl4) [136]. This interesting model reproduces the progressive stages of human NAFLD, from simple steatosis, to inflammation, fibrosis, and HCC. In addition, the model replicates gene expression and immune abnormalities of human disease [136]. Nevertheless, previous reports showed that CCl4 is a potent hepatotoxin that can cause genotoxicity and oxidative DNA damage in rats [137]. Numerous genetically modified mice that are susceptible to NASH and HCC development have been described but most of them do not resemble the human progression of disease. HFD-fed MUP-uPA and DIAMOND mice develop human NASH-like disease and almost all of them progress to HCC [12]. However, in these models HCC development is significantly slower than in toxins/diet-based models. Moreover, the mutational landscape of these mice differs significantly from mouse to mouse [93].

Aim of research is overcoming this gap in order to identify a model that best replicates the several aspects of NASH-driven HCC, because studies that are performed with inappropriate models generate misleading results that delay progress in this field.

The understanding of the multiple molecular mechanisms involved in fibrogenesis make it possible to identify various therapeutic targets including cytokines, chemokines, HSCs, Hedgehog pathway signaling and other potential targets, for the purpose of reversing the fibrosis process. A better understanding of the underlying pathophysiological mechanisms could also be useful for identify non-invasive biomarkers of NASH and fibrosis because liver biopsy as the diagnostic “gold standard” it is not without risk. Moreover, imaging techniques can detect steatosis but not steatohepatitis.

The results of preclinical experiments are promising, as they have shown that it is possible to reverse the fibrogenesis process. However, it is not that simple to verify in humans. Furthermore, it is desiderable that future research establishes whether a reduction of fibrosis in patients with NASH is effective in HCC prevention [138].

## Figures and Tables

**Figure 1 ijms-20-01745-f001:**
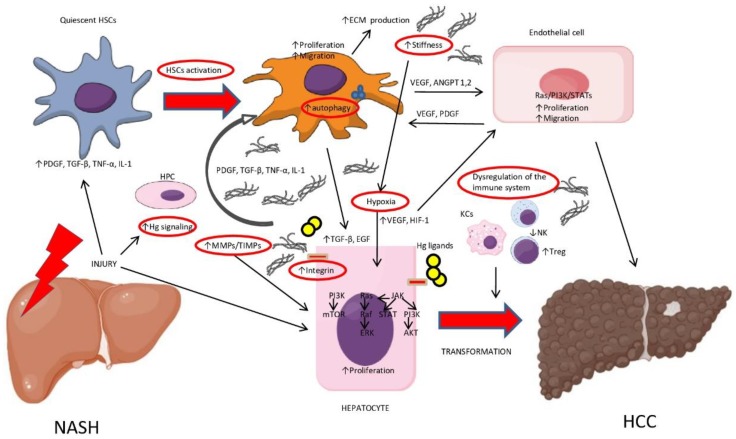
Fibrosis-dependent mechanisms of hepatocarcinogenesis in non-alcoholic steatohepatitis (NASH). Abbreviations: Hh, Hedgehog; PDGF, platelet derived growth factor; TGF-β, transforming growth factors-β; TNF-α, tumor necrosis factor-α; IL-1, interleukin-1; HSCs, hepatic stellate cells; ECM, extracellular matrix; PI3K, phosphatidylinositol 3-kinase; EGF, epidermal growth factor; MMPs, matrix metalloproteinases; TIMPs, tissue inhibitor of metalloproteases; VEGF, vascular endothelial growth factor; HIF-1, hypoxia-inducible factor 1; ANGPT, angiopoietin; HPCs, hepatic progenitor cells; KCs, Kupffer Cells.

**Table 1 ijms-20-01745-t001:** Main mechanisms involved in NASH progression and hepatocellular carcinoma (HCC) development.

Factor	Mechanism	Biological Effects
Inflammation	↑ PDGF, TGF-β, TNF-α, IL-1 and chemokines	HSCs and KCs activation
Activated HSCs	↑ Type I and III collagens, fibronectins, laminins, fibrinogen secretion ↑ TGF-β_1_, TGF-α, PDGF, FGF, EGF, VEGF, ANGPT 1-2, HGF, SDF-1, Wnt ligands, IL-6, EPM	↓ MMPs↓ and ECM turnover ↑ Stiffness ↑ Hypoxia ↓ NK cell functions ↑ Integrin signaling ↑ HSCs activation ↑ Angiogenesis
↑ Stiffness	↑ Collagen secretion Activation of ERK, PKB/Akt, STAT3, integrin β1/GSK-3β/β-catenin; β1-integrin/FAK and β1-integrin/PI3K/Akt signaling pathways Activation of Hippo-YAP/TAZ signaling pathway and integrin β1/Akt/mTOR/SOX2 signaling pathway	↑ Cell proliferation, motility and tumor metastasis ↓ Apoptosis Stemness dysregulation ↓ NK cell functions
Integrins	↑ Hedgehog signaling, PI3K, MAPK pathways	↑ Proliferation, survival ↑ Angiogenesis
MMPs/TIMPs imbalance	↑ Release of growth factors, ligands and citokines from the ECM ↑ TGF-β signaling ↓ Alpha-1 proteinase inhibitor ↓ SDF-1/PI3K/AKT signaling	↓ ECM turnover ↑ HSCs activation Progression of hepatocytes from dysplasia to HCC ↑ Migration ↑ Invasion ↓ Apoptosis
Hypoxia	Activation of PI3K-Akt and MAPK pathways ↑ VEGF and HIF-1 ↑ TUFT1 →Ca2 /PI3K/AKT pathway	KCs activation ↑ Angiogenesis, ↑ EMT ↑ Tumor growth and metastasis
Hedgehog (Hg)	↑ TGF-β1 and HSCs activation	NASH progression ↑ Proliferation ↑ Migration and invasion of HCC cells
HPCs and ductular reaction (DR)	↑ Hg, Wnt signaling and Notch pathways ↑ PDGF, TGF-β, and HSCs activation	NASH progression
Autophagy	PI3K/Akt/mTOR and AMPK pathway Mitochondrial ROS- NF-kB-IL1α/β pathways	±Cellular proliferation and apoptosis
Kupffer Cells (KCs)	↑ TGF-β, TNF-α, CCL2, ROS Activation of NLRP3 →IL-1β, IL-18 and IL-6	NASH progression ↑ Proliferation
↓ NK cells	↓ IFNγ ↓ Apoptosis of HSCs	↑ ECM deposition ↓ Tumor surveillance
NKT cells	IL-4, IFN-γ, and TNF-α Regulate HSCs activation and Hh signaling	Anti-tumor role or tumor tolerance
↑ Treg cells	↓ NK and CD8+ T cells	↓ Tumor surveillance

Hh, Hedgehog; PDGF, platelet derived growth factor; TGF-β, transforming growth factors-β; TNF-α, tumor necrosis factor-α; IL-1, interleukin-1; HSCs, hepatic stellate cells; ECM, extracellular matrix; PI3K, phosphatidylinositol 3-kinase; EGF, epidermal growth factor; MMPs, matrix metalloproteinases; TIMPs, tissue inhibitor of metalloproteases; VEGF, vascular endothelial growth factor; HIF-1, hypoxia-inducible factor 1; ANGPT, angiopoietin; HPCs, Hepatic progenitor cells; EPM, epimorphin.

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
