# Peer review of "Recent Insight into the Role of Fibrosis in Nonalcoholic Steatohepatitis-Related Hepatocellular Carcinoma"

_ijms, 2019, doi:10.3390/ijms20071745_

Round 1

Reviewer 1 Report

The review by Sircana and co-workers deals with the role of hepatic fibrosis in the pathogenesis of hepatocellular carcinoma  (HCC) associated with non-alcoholic fatty liver disease (NAFLD). 

In their work the authors discuss the possible implication in HCC pathogenesis of integrin, hepatic stellate cells (HSCs) activation, Hedgehog signalling and alteration of innate immunity, but forget to mention the implication of hepatic progenitor cells, that are greatly expanded in NAFLD, as well as the contribution of adaptive immunity.   On this later respect, the immunosuppressive action of IgA-producing plasma cells is mentioned in page 6, but the inclusion in innate immunity section is incorrect.   Furthermore, the discussion of the role of innate in immunity NAFLD related HCC should consider the modulation of inflammatory mechanisms as well as the implication of NKT cells.

Additional information on these topics can be found in the following papers.

1)      Ringelhan M, et al. The immunology of hepatocellular carcinoma.  Nat Immunol. 2018;19:222-232. doi: 10.1038/s41590-018-0044-z

2)      Park EJ, et al. Dietary and genetic obesity promote liver inflammation and tumorigenesis by enhancing IL-6 and TNF expression. Cell. 2010;140:197-208. doi: 10.1016/j.cell.2009.12.052.

3)      De Minicis S, et al. HCC development is associated to peripheral insulin resistance in a mouse model of NASH. PLoS One. 2014;9:e97136. doi: 10.1371/journal.pone.0097136

4)      Nakagawa H, et al. ER stress cooperates with hypernutrition to trigger TNF-dependent spontaneous HCC development. Cancer Cell. 2014;26:331-343. doi: 10.1016/j.ccr.2014.07.001

5)      Ma C, et al. NAFLD causes selective CD4(+) T lymphocyte loss and promotes hepatocarcinogenesis. Nature. 2016;531:253-257. doi: 10.1038/nature16969.

6)      Shalapour S, et al. Inflammation-induced IgA+ cells dismantle anti-liver cancer immunity. Nature. 2017;551:340-345. doi: 10.1038/nature24302.

7)      Wolf MJ, et al.  Metabolic activation of intrahepatic CD8+ T cells and NKT cells causes nonalcoholic steatohepatitis and liver cancer via cross-talk with hepatocytes.  Cancer Cell. 2014;26:549-564. doi: 10.1016/j.ccell.2014.09.003.

8)      Grohmann M, et al.  Obesity Drives STAT-1-Dependent NASH and STAT-3-Dependent HCC. Cell. 2018;175:1289-1306.e20. doi: 10.1016/j.cell.2018.09.053.

Author Response

-We would like to thank the reviewer for the valuable material recommended to us to improve the quality of the manuscript.
-As suggested, in the paragraph ‘Dysregulation of the immune system’ we considered both innate and adaptive immune system including the modulation of inflammatory mechanisms and the implication of NKT cells.
-At page 6 we discussed the implication of hepatic progenitor cells in NASH.
-Native English speakers evaluated the entire manuscript.

Reviewer 2 Report

         In this review entitled “Recent Insights into the Role of Fibrosis in Nonalcoholic Steatohepatitis-Related Hepatocellular Carcinoma” The authors provide recent advances in studying the pathophysiological mechanisms linking liver fibrogenesis to hepatocellular carcinoma (HCC) in nonalcoholic steatohepatitis (NASH). They discussed the potential mechanisms of fibrosis-dependent hepatocarcinogenesis. These include the extracellular matrix (ECM)-mediated integrin signaling, hepatic stellate cells (HSCs) activation, hypoxia, Hedgehog signaling, autophagy, crosstalk between hepatocytes, HSCs and the ECM, and change of innate immunity. They also discuss the potential therapeutic targets based on the latest findings for the mechanistic links between liver fibrogenesis and HCC in NASH. Overall this review is well organized and written, and provides a critical appraisal of previous studies related to fibrosis in NASH-related HCC. A few suggestions as below:

1.    Although the authors addressed a greater risk of HCC in NASH-cirrhosis cohorts than in NASH without fibrosis/cirrhosis, it was weak that the pathogenesis of NASH towards HCC without fibrosis/cirrhosis. The authors may discuss recent advances in researching the crosstalk between NASH and HCC.

2.    The authors have nicely discussed the role of fibrosis in NASH-related HCC. I am just wondering whether the authors could indicate the shortage for the current studies and what the further study is attempting to accomplish.

3.    It would be better if the authors could also discuss some key and expanding pathogenesis of NASH in the development of fibrosis.

             4.    A murine NASH model with rapid progression of NASH, fibrosis and HCC has been established (J Hepatol. 2018, 69:385-395). The authors might expand the discussion for the pathophysiological mechanisms in NASH-related HCC in an animal model.

Author Response

1. Although the authors addressed a greater risk of HCC in NASH-cirrhosis cohorts than in NASH without fibrosis/cirrhosis, it was weak that the pathogenesis of NASH towards HCC without fibrosis/cirrhosis. The authors may discuss recent advances in researching the crosstalk between NASH and HCC.
Reply: In the paragraph ‘Crosstalk Between NASH and Hepatocellular Carcinoma’ we discussed the bidirectional relationship between tumor cells and their surrounding microenvironment in NASH.

2. The authors have nicely discussed the role of fibrosis in NASH-related HCC. I am just wondering whether the authors could indicate the shortage for the current studies and what the further study is attempting to accomplish.
Reply: In the paragraph conclusions and discussion we examine the limits of  current studies and the aim of future research.

3. It would be better if the authors could also discuss some key and expanding pathogenesis of NASH in the development of fibrosis.
Reply: As suggested we added a paragraph entitled ‘Pathophysiological Mechanisms of NAFLD Progression’.

4. A murine NASH model with rapid progression of NASH, fibrosis and HCC has been established (J Hepatol. 2018, 69:385-395). The authors might expand the discussion for the pathophysiological mechanisms in NASH-related HCC in an animal model.
Reply: In the ‘conclusions and discussion’ section we have briefly discussed the most frequently used mouse models and the limits of each of them.
-Native English speakers evaluated the entire manuscript.

Reviewer 3 Report

In this report, the authors summarize recent studies linking pathological mechanisms associated with NASH fibrosis and HCC. The topic is of timely importance and the authors cover a lot of material in the review. The content and quality of the paper would benefit from consideration of the following suggestions:

Tables summarizing main findings from each pathway category would be useful.

A critical discussion of the research findings, as opposed to a summary. would add intellectual value to the review article. Likewise, a discussion of remaining gaps, future research directions, and critical next steps would be valuable.

Author Response

Tables summarizing main findings from each pathway category would be useful.
Reply: As suggested, we added a table we added a table that summarizes pathophysiological mechanisms linking liver fibrogenesis to HCC in NASH.

A critical discussion of the research findings, as opposed to a summary. would add intellectual value to the review article. Likewise, a discussion of remaining gaps, future research directions, and critical next steps would be valuable.
Reply: In the ‘conclusions and discussion’ paragraph we examine the limits of  current studies and the aim of future research.

-Native English speakers evaluated the entire manuscript.

Round 2

Reviewer 1 Report

None